# Assessment of In Silico and In Vitro Selpercatinib Metabolic Stability in Human Liver Microsomes Using a Validated LC-MS/MS Method

**DOI:** 10.3390/molecules28062618

**Published:** 2023-03-14

**Authors:** Mohamed W. Attwa, Haitham AlRabiah, Gamal A.E. Mostafa, Ahmed H. Bakheit, Adnan A. Kadi

**Affiliations:** Department of Pharmaceutical Chemistry, College of Pharmacy, King Saud University, Riyadh 11451, Saudi Arabia

**Keywords:** selpercatinib, metabolic stability, in vitro half-life, intrinsic clearance, LC-MS/MS, WhichP450 module

## Abstract

Selpercatinib (SLP; brand name Retevmo^®^) is a selective and potent RE arranged during transfection (RET) inhibitor. On 21 September 2022, the FDA granted regular approval to SLP (Retevmo, Eli Lilly, and Company). It is considered the only and first RET inhibitor for adults with metastatic or locally advanced solid tumors with RET gene fusion. In the current experiment, a highly specific, sensitive, and fast liquid chromatography tandem mass spectrometry (LC-MS/MS) method for quantifying SLP in human liver microsomes (HLMs) was developed and applied to the metabolic stability evaluation of SLP. The LC-MS/MS method was validated following the bioanalytical methodology validation guidelines outlined by the FDA (linearity, selectivity, matrix effect, accuracy, precision, carryover, and extraction recovery). SLP was detected by a triple quadrupole detector (TQD) using a positive ESI source and multiple reaction monitoring (MRM) mode for mass spectrometric analysis and estimation of analytes ions. The IS-normalized matrix effect and extraction recovery were acceptable according to the FDA guidelines for the bioanalysis of SLP. The SLP calibration standards were linear from 1 to 3000 ng/mL HLMs matrix, with a regression equation (y = 1.7298x + 3.62941) and coefficient of variation (r^2^ = 0.9949). The intra-batch and inter-batch precision and accuracy of the developed LC-MS/MS method were −6.56–5.22% and 5.08–3.15%, respectively. SLP and filgotinib (FLG) (internal standard; IS) were chromatographically separated using a Luna 3 µm PFP (2) stationary phase (150 × 4.6 mm) with an isocratic mobile phase at 23 ± 1 °C. The limit of quantification (LOQ) was 0.78 ng/mL, revealing the LC-MS/MS method sensitivity. The intrinsic clearance and in vitro t_1/2_ (metabolic stability) of SLP in the HLMs matrix were 34 mL/min/kg and 23.82 min, respectively, which proposed an intermediate metabolic clearance rate of SLP, confirming the great value of this type of kinetic experiment for more accurate metabolic stability predictions. The literature review approved that the established LC-MS/MS method is the first developed and reported method for quantifying SLP metabolic stability.

## 1. Introduction

Cancer is considered the main worldwide cause of death and is described by the abnormal growth of cells with the possibility of spreading to various parts of the body. It occurs because of the damage to genes, which are responsible for controlling cellular functions and could permit a cell to be malignant [1]. Molecular targeting approaches have been lately utilized for cancer treatment depending on the tumor oncogenes and suppressor genes that are contributing to the development of human cancers [2]. An example of a genetic alteration that is currently cured with various types of certain small molecules is rearranged during transfection (RET), which is documented in various types of cancer, such as colon cancer, non-small cell lung cancer (NSCLC), medullary thyroid cancer, papillary thyroid cancer, and other types of solid tumors [3]. The progression of tumors in NSCLC strongly accounts for RET activity. Certain mutations can lead to the activation of the tyrosine kinase signal transduction pathway and endless autophosphorylation that generates an alteration in the tyrosine kinase specificity, which leads to tumor development. Numerous drugs have been established to attack this process and confirm that tyrosine kinases (TKIs) are stopped [4].

Selpercatinib (Retevmo^®^; SLP) is a highly selective RET kinase inhibitor with activity against different RET fusions, activating RET mutations, and brain metastases [5]. On September 21, 2022, SLP (Retevmo^®^, Eli Lilly and Company) was granted accelerated approval by the Food and Drug Administration (FDA) in the United States for patients with either locally advanced or metastatic solid tumors involving metastatic RET fusion-positive NSCLCs along with RET fusion-positive thyroid cancers and RET-mutant medullary thyroid cancers that have proceeded on or after preceding systemic treatment or who have no suitable other treatment choices [6]. SLP exhibit better results for patients with RET fusion-positive cancer due to better potency and fewer side effects. SLP side effects included a grade 1 or 2 rise in alanine transferase levels, elevated serum bilirubin, or constipation. There are no reported treatment-related toxicities for SLP [7]. Filgotinib (FLG) was utilized as an internal standard in SLP analytical method development (Figure 1).

Upon literature review, two LC-MS/MS methods were reported for quantifying SLP in mouse plasma or rat plasma that were applied for SLP application to pharmacokinetic studies in mice or rats [8,9]. In the two published analytical methods, calibrations were done from 1 to 2000 ng/mL or from 2 to 2000 ng/mL using a 1/x least squares quadratic regression (linearity model) based on calibration levels using a gradient mobile phase with no resolution between analytes. Additionally, both reported articles used single reaction monitoring (SRM), which is less sensitive than multiple reaction monitoring (MRM). In the current analytical method, we established an LC-MS/MS methodology for quantifying SLP in HLMs that was applied for metabolic stability evaluation of SLP. The established LC-MS/MS method is characterized by the use of isocratic mobile phase at a lower flow rate (0.4 mL) and a shorter run time (3 min), which is considered more precise and consumes less organic solvent (green chemistry) in comparison with the reported method [8]. Furthermore, the established calibration curve exhibited a linearity in the range of 1–3000 ng/mL that revealed a wider range than the previously reported calibration model [8]. Another advantage of the current study is the quantification of SLP using MRM transitions (*m*/*z* 525→122 and 525→59) which are more accurate and sensitive compared to the SRM transition (*m*/*z* 525→122) in the other two previously reported methods [8,9]. The current work is considered the first analytical method to evaluate SLP metabolic stability in HLMs.

Before starting the LC-MS/MS development for the application of metabolic stability evaluation, SLP should be checked for its metabolic lability to Cyp450 enzymes using the in silico software P450 metabolic model of the StarDrop software package [10]. Thus, an in silico metabolic assessment was done for SLP metabolic lability. However, there are no reported analytical methods, including LC-MS/MS methodology, for the SLP estimation in HLM’s matrix. So, the current analytical method (LC-MS/MS) was established and validated in HLM’s matrix for quantifying SLP using calibration standards and unknowns. The results of metabolic stability are included in the calculation of two parameters [the intrinsic clearance (Cl_int_) and the in vitro half-life (t_1/2_)] of SLP [11] using an ‘in vitro t_1/2_′ approach (‘well-stirred’ model) [12,13] as it is considered the most utilized model in various metabolism experiments owing to its ease. In silico software data were utilized as a clue for the in vitro incubation experiments and generated data about the rate of SLP metabolism that provided a proposed figure for SLP in vivo bioavailability [12,13]. If a drug exhibits high metabolic clearance and a short in vitro t_1/2_, it will have a fast duration of action and low in vivo bioavailability [14,15,16,17].

## 2. Results and Discussion

### 2.1. In Silico SLP Metabolic Stability

The exhibited metabolic landscape represents the degree of metabolic lability of different active sites inside the SLP chemical structure towards the major metabolizing enzyme (CYP3A4), as exhibited in the pie chart, so as to give a full idea about its metabolic rate [18,19,20]. The composite site lability (CSL; 0.9797) proposed a high SLP metabolic rate; consequently, the established LC-MS/MS was applied for the SLP metabolic stability calculation (Figure 2). The lability of metabolism is classified from labile (highest) to stable (lowest) according to the CSL value. The results indicated that C1 and C3 of the methylpropoxy group, C38 of the methoxy group, and C30 of the methyl group linked to the pyridine group are labile, while C4 of the methylpropoxy group is moderately labile. The obtained outcomes for the metabolic landscape confirmed that the methylpropoxy group (C1, C3, and C4) is the chief cause of SLP metabolic lability, as revealed by the CSL outlined in Figure 2 (value of 0.9797, indicating high lability to metabolism), which indicated the great value of establishing an analytical method that was applied for practical estimation of SLP metabolic stability and gave the same results as in silico software. From the previous results, the P450 metabolism model could be utilized for a preliminary test for the metabolic lability of drugs in a proficient way that matched with the practical metabolic incubation work (see below).

### 2.2. LC–MS/MS Method

The FLG drug was selected as the internal calibration standard (IS) in the established LC-MS/MS method for quantifying SLP in the HLMs due to three reasons: first, both analytes (SLP and FLG) could be extracted from the HLM’s matrix using the protein precipitation extraction method with a high recovery rate for SLP (100.8 ± 3.14%) and FLG (102.34 ± 2.86%), second, the chromatographic peaks of FLC (1.1 min) and SLP (2.1 min) were eluted in a short run time (3 min) with a good separation, and third, there is no published case for the simultaneous use of both drugs (SLP and FLG) in the same prescription for a patient. So, the developed LC-MS/MS method could be used for absorption, distribution, metabolism, and excretion (ADME) studies of SLP without the fear of interference from co-administered FLG.

The LC-MS/MS method is characterized by a good sensitivity of 1 ng/mL of SLP at 2.1 min (Figure 3A). Figure 3B illustrates the reproducibility of the LC-MS/MS method by showing the MRM chromatograms of SLP calibration levels (from 1 ng/mL to 3000 ng/mL) that revealed the same retention times for all calibration standards of SLP. No observed carryover effect was found for SLP in blank HLMs plus FLG MRM chromatograms (Figure 3C).

### 2.3. Validation Parameters

#### 2.3.1. Specificity

The LC-MS/MS method exhibited good specificity, which was confirmed by the optimum separation and resolution of the eluted analytes peaks of SLP and FLG, as shown in Figure 3. Additionally, there were no chromatographic peaks from the HLM’s matrix constituents in the MRM chromatogram that interfered with the SLP chromatographic peak at its retention time due to the use of the MRM analyzer mode (Figure 3C). After multiple injections of SLP standards and injections of blank samples, no carry-over effect was detected in the negative or positive control chromatograms (Figure 3C).

#### 2.3.2. Sensitivity and Linearity

The LC-MS/MS method exhibited good linearity in the range of 1–3000 ng/mL that was revealed statistically utilizing the linear regression equation (y = 1.7298x + 3.62941; r^2^ = 0.9949) by injecting eleven different SLP levels in HLM’s matrix and then back calculating as unknowns. No weighting of the calibration line was applied. The RSD values for the six replicates (eleven standards) were <2.15% (Table 1). The LOD and LOQ were 0.26 ng/mL and 0.78 ng/mL, respectively.

#### 2.3.3. Precision and Accuracy

The LC-MS/MS method exhibited optimum precision and accuracy either intra-batch or inter-batch, as revealed statistically by injecting twelve replicates (quality controls) in the same day and six replicates (QCs) in three consecutive days, respectively. The results were in the acceptable range as stated by the FDA guidelines [21]. The intra- and inter-batch accuracy and precision of the LC-MS/MS were −5.08–3.15% and −6.56–5.22%, respectively (Table 2).

#### 2.3.4. Matrix Effects and Extraction Recovery

The extraction recovery proficiency of the used extraction method (protein precipitation) using ACN for both analytes (SLP and FLG) was approved by injecting six replicates (QCs) in HLM’s matrix and comparing it with quality control prepared in the mobile phase. The results revealed high recovery percents for SLP (99.55 ± 3.92 and RSD < 3.93%) and FLG (102.32 ± 3.62%). The absence of matrix influence on the ionization efficiency of SLP or FLG was verified by injecting two sets of HLM matrices. Set 1 of the HLMs matrix was spiked with the SLP LQC (3 ng/mL) and FLG (1000 ng/mL), while Set 2 was performed using the mobile phase in place of the HLMs matrix. Using Equation (1), the HLMs containing SLP and FLG showed a matrix effect of 99.67 ± 2.04% and 102.24 ± 2.26%, respectively. Using Equation (2), the IS normalized ME was 0.97.5 and was in the accepted range following FDA guidelines. Consequently, these data revealed that the HLM’s matrix showed no obvious effect on the ionization efficiency of either SLP or FLG.
(1)Matrix effect of SLP or FLG=mean peak area ratio Set 1Set 2×100
(2)IS normalized ME=matrix effect of SLPmatrix effect of FLG (IS)

### 2.4. Metabolic Stability

In metabolic stability experiments, the concentration of the analyte (SLP) used should be 1 µM/mL of the incubation mixture to be less than the Michaelis–Menten constant to establish a linear relationship between the time of incubation and the SLP metabolic rate. Additionally, 1 mg of microsomal protein (HLMs) should be used in the HLMs incubation mixture to escape the nonspecific protein binding. The concentration of SLP was calculated using the regression equation of a concurrently injected SLP calibration standard. The first SLP metabolic stability curve was established by plotting the stopped time points (*x*-axis) from 0 to 70 min against the percentage remaining of SLP concentration compared to the zero-time concentration (*y*-axis) (Figure 4A). The linear part (0–30 min) of the established curve was chosen to establish another natural logarithmic curve of incubation time points (0–30 min) against the natural logarithm (Ln) of the % SLP remaining (Figure 4B). The slope of the second metabolic curve (0.0291) reflected the SLP metabolic rate constant, while its regression equation (y = −0.0291x + 4.6221 with R^2^ = 0.9971) was used for computing SLP in vitro t_1/2_ (Table 3). From the previous regression equation, the slope was 0.0291 and was chosen to calculate the in vitro t_1/2_, whereas in vitro t_1/2_ = ln2/slope, so in vitro t_1/2_ was 23.82 min. SLP Cl_int_ was calculated following the in vitro t_1/2_ method, so the Cl_int_ of SLP was 34 mL/min/kg using a value of 26 g for liver tissue per kilogram of body weight and of 45 mg of microsomal protein per gram of liver tissue Equation (3).
(3)$${CL}_{int,}=\frac{0.693}{{in\;vitro\;t}_{½}}\times \frac{mL\;incubation}{mg\;microsomes}\times \frac{mg\;microsomal\;proteins}{g\;liver}\times \frac{g\;liver}{Kg\;b.w.}\;{CL}_{int,}=\frac{0.693}{23.82}\times \frac{1}{1}\times \frac{45}{1}\times \frac{26}{1}$$

Metabolic stability, which gives an estimate of the susceptibility to biotransformation, can be computed from HLM incubation-based half-life and intrinsic clearance calculations. Following the scoring proposed by McNaney et al. [22], SLP is estimated to be an intermediate clearance compound, which highlights the importance of this type of kinetic study for more accurate metabolic susceptibility predictions. By utilizing other software (the simulation and Cloe PK software), these outcomes could also be utilized to predict SLP in vivo pharmacokinetics [23].

## 3. Material and Methods

### 3.1. Materials and Instruments

All solvents used in the current experiment were of HPLC grade. All chemicals and reference powders were of AR grade. SLP reference powder (synonyms: LOXO-292; Cat. No.: HY-114370 at purity: 99.87%) and filgotinib reference powder (synonyms: GLPG0634; Cat. No.: HY-18300 at purity: 99.37%) were procured from MedChem Express (Princeton, NJ, USA). HLMs (20 mg/mL; M0567), formic acid, ammonium formate, and acetonitrile were purchased from Sigma-Aldrich Company (St. Louis, MO, USA). HLMs were kept at −70 °C until use. In-house Milli-Q Plus purification equipment, purchased from Millipore Company (Billerica, MA, USA), generates the required water at HPLC grade. The LC-MS/MS (UPLC-TQD MS) system was used for chromatographic separation of analytes peaks and mass spectrometric detection of separated ion peaks. UPLC (serial number: H10UPH], electrospray ionization source (ESI), and Acquity TQD MS (serial number: QBB1203) were managed by MassLynx 4.1 software. The mass tuning was assisted by the IntelliStart^®^ module in MassLynx 4.1 software (version 4.1, SCN 805). The processing, acquisition of data, and reporting of outcomes were automatically done by using ‘QuanLynx’ module, which is considered one of the application managers in the MassLynx 4.1 software package. A nitrogen generator from the Peak Scientific Company (Renfrewshire, Scotland, UK) was used for supplying nitrogen gas (desolvation gas), and a Sogevac^®^ vacuum pump (Murrysville, PA, USA) was used for establishing vacuum. Argon gas of 99.999% purity was obtained from a local supplier to be used as a collision gas for generating fragments from the analyte parent ions inside the second quadrupole of the TQD analyzer (collision cell).

### 3.2. In Silico SLP Metabolic Stability Assessment

In silico SLP metabolic lability evaluation was performed using the StarDrop software package (WhichP450 metabolic model) from Optibrium Ltd., Cambridge, MA, USA. The results were summarized as a value (CSL) revealing the metabolic lability degree [24,25,26] that was calculated by combining the atom labilities of individual atoms [24,25,26] using Equation (4):(4)CSL=ktotal(ktotal+kw)
as *kw* is the water formation rate constant.

CSL is considered a very essential parameter in proposing the metabolism rate of SLP before developing the practical experiments (in vitro incubations) to reveal the value of the current experiment and to save time by not running unnecessary experiments.

### 3.3. LC-MS/MS Analytical Methodology

#### 3.3.1. Liquid Chromatographic

The LC chromatographic parameters controlling the resolution of the two analytes (SLP and FLG), such as the composition and pH of the mobile phase and the type of stationary phase, were optimized. The mobile phase consisted of an organic part (55% ACN) and an aqueous part [45% aqueous solution (0.1% 10 mM ammonium formate in H_2_O)] at a flow rate of 0.4 mL/min. The aqueous portion of the mobile phase was a 10 mM NH_4_COOH solution in water, and its pH was fixed at 4.8 using a few drops of formic acid. Elevating the pH value (more than 4.8) caused an unnecessary long running time and tailing of the SLP and FLG chromatographic peaks. The organic portion of the mobile phase was ACN. Elevating the ACN % of generated overlapped chromatographic peaks and a poor resolution while reducing the ACN % generated long running time. Various types of stationary phases were experienced, such as polar columns (HILIC columns). Though neither SLP nor FLG were retained, the optimum outcomes were attained through the use of a Luna 3 µm PFP(2) column (150 × 4.6 mm; Part No.: 00F-4447-E0). The column temperature was kept at room temperature at 23 ± 1 °C. Run time and injection volume were 3 min and 5 µL, respectively. The autosampler temperature was kept at 5 °C to increase the stability of samples. Various trials were done to choose the perfect chromatographic condition for resolution, extraction, and analysis of SLP and FLG analytes peaks in a good shape and with a short retention time, as mentioned in Table 4.

#### 3.3.2. Mass Spectrometry

The TQD mass spectrometry parameters were optimized to get good sensitivity and ionization for the two analytes (SLP and FLG). The quantification of SLP (C_29_H_31_N_7_O_3_) and FLG (C_21_H_23_N_5_O_3_S) chromatographic peaks was performed using TQD MS after the generation of ions from a positive ESI ionization source. The tuning for SLP and FLG was performed automatically using IntelliStart^®^ software (Version 4.1, SCN 805), which was fixed manually in fluidics and LC (combined mode) so as to elevate the intensity and selectivity of the chromatographic peaks of the two analytes (SLP and FLG). The flow rate of argon gas with a purity of 99.999% was adjusted to 0.14 mL/min and utilized as a collision gas inside the collision cell. Nitrogen at a flow rate of 650 L/H (350 °C) was used as a drying gas that was supplied by a nitrogen generator. The flow rate of the cone gas was fixed at 100 L/H. MRM mass analyzer detection mode was used as a mode of mass analyzer for quantifying the two analytes (SLP and FLG) in order to increase the selectivity and sensitivity of the established LC-MS/MS analytical method. The extractor voltage, capillary voltage, and RF lens were set at 3.0 (V), 4 (kV), and 0.1 (V), respectively. The source temperature was adjusted to 150 °C. The dwell time for SLP and FLG mass transitions was 0.025 s. FLG and SLP chromatographic peaks were eluted at Rt: 1.1 min (Figure 5A) and Rt: 2.1 min (Figure 5B), respectively, that were quantified through the use of MRM mode transitions. All MRM optimized parameters of SLP and FLG (IS) using IntelliStart^®^ software are listed in Table 5.

### 3.4. SLP Working Solutions

SLP and FLG are soluble in DMSO at 62.5 mg/mL (118.91 mM; Need ultrasonic) and 25 mg/mL (58.75 mM; Need ultrasonic), respectively. Therefore, SLP and FLG stock solutions were made in DMSO at 1 mg/mL followed by sequential dilution using mobile phase to prepare SLP-working solution 1 (WK1: 100 µg/mL), SLP WK2 (10 µg/mL), SLP WK3 (1 µg/mL), and FLG WK3 (10 µg/mL), respectively.

### 3.5. SLP Calibration Standards

DMSO quenched metabolic incubations, even at low concentrations (0.2%) [27]. So, DMSO was used for the deactivation of HLMs at 2% concentration with slight warming as heat stopped microsome activity at 50 °C for 5 min [28,29]. The deactivation of HLMs was performed to avoid any effect on the SLP calibration levels during the validation steps. After deactivation, the preparation of HLMs matrix was achieved at a concentration of 1 mg protein/mL by diluting 30 µL of HLMs (deactivated) to 1 mL with 0.1 M sodium phosphate buffer (pH 7.4) containing 1 mM NADPH. SLP calibration standards were performed by sequential dilution of SLP WK2 (10 µg/mL) and SLP WK3 (1 µg/mL), with the HLM matrix revealing eleven levels: 1, 3, 15, 50, 100, 300, 500, 900, 1500, 2400, and 3000 ng/mL. The HLM matrix volume was maintained at not less than 90% of the total volume of the prepared solution so as to decrease the effect of dilution at the time of real sample analysis. These SLP standards were used for constructing a calibration curve. Four SLP standards were chosen as QCs for the validation procedure: 1 ng/mL (LLQC), 3 ng/mL (LQC), 900 ng/mL (MQC), and 2400 ng/mL (HQC). One hundred microliters of 10 µg/mL FLG were added as the IS to 1 mL of calibration levels and QCs that contain deactivated HLMs.

The protein precipitation was used as an extraction method for the two analytes (SLP and FLG) from the HLM’s incubation matrix in three steps. First, 2 mL of ACN was added to the SLP calibration standards or unknown samples. Second, centrifugation of all standards and samples at 14,000 rpm for 12 min (thermostated at 4 °C) to clarify supernatants by precipitating and collecting proteins. Third, 1 mL of the supernatant was passed through a syringe filter (0.22 µm) into HPLC vials to verify the suitability of samples to be injected into the LC-MS/MS system. Five µL were injected into the UPLC-TQD MS system. Negative control (HLMs matrix) and positive control (HLMs matrix plus IS) were prepared following the three steps mentioned above to confirm the absence from HLMs matrix at the elution times of both analytes (SLP and FLG). A SLP calibration curve was established by plotting the peak area ratio from SLP to FLG (*y*-axis) versus SLP nominal values (*x*-axis). The linear regression equation (y = ax + b; r^2^) was utilized for confirming the linearity of the developed LC-MS/MS method.

### 3.6. Method Validation

The validation of the established LC-MS/MS method was performed through linearity, sensitivity, specificity, precision, accuracy, extraction recovery, and matrix effect validation parameters. The bioanalytical method validation steps for the established UPLC-TQD MS method were performed according to FDA general regulations [21].

#### 3.6.1. Specificity

The specificity of the LC-MS/MS method was assessed by analyzing six blank HLM matrix samples after extraction using the same previously mentioned procedure. Then, these extracts were injected into the LC-MS/MS system and tested for any interference chromatographic peaks at SLP or FLG retention times, matching the chromatogram of spiked HLMs matrix samples with the two analytes (SLP and FLG). MRM mode was used so as to decrease the carryover effects of the two analytes (SLP and FLG) in the mass analyzer.

#### 3.6.2. Linearity and Sensitivity

Linearity and sensitivity of the established LC-MS/MS method were assessed by injecting 12 calibration curves (eleven standards) of SLP in HLM’s matrix, then back calculating as unknowns. The linearity of the UPLC-TQD MS method was approved statistically using the least-squared statistical method (y = ax + b). The LOQ and LOD were calculated as stated in the pharmacopeia utilizing the slope of the constructed calibration curve and the standard deviation (SD) of the intercept as LOQ = 10* S SD of intercept/slope and LOD = 3.3 ∗ SD of intercept/slope.

#### 3.6.3. Accuracy and Precision

The precision and accuracy of the LC-MS/MS method were approved intra-batch and inter-batch by injecting twelve replicates of SLP quality controls in the same day and six replicates (QCs) in three following days, respectively, following USFDA guidelines. For expressing the accuracy and precision of the current LC-MS/MS method, % error and % relative standard deviation (% RSD) were used, respectively. Percentages RSD = (SD/Mean) × 100 and % error = [(average calculated conc. − proposed conc.)/proposed conc.] × 100].

#### 3.6.4. Matrix Effect and Extraction Recovery

The matrix effects and recovery of SLP in HLM’s matrix were tested using QC samples. The extraction and recovery efficiency of the protein precipitation methodology using ACN for both analytes (SLP and FLG) was approved by injecting six replicates (QCs) in HLM’s matrix (B) and comparing them with QCs prepared in the mobile phase (A). The % recovery is the ratio of B/A × 100. The lack of matrix effect on the ionization efficiency of both analytes (SLP or FLG) was confirmed by injecting two groups of HLM matrix samples. Group 1 of the HLMs matrix was spiked with the SLP LQC (3 ng/mL) and FLG (1000 ng/mL), while group 2 was done by using the mobile phase instead of the HLMs matrix. The matrix effects (ME) for SLP and FLG were computed using Equation (1). The IS-normalized ME was computed using Equation (2).

### 3.7. SLP Metabolic Stability

The concentration of SLP in the incubation mixture was computed using the regression of a simultaneous injected SLP calibration standard. The first SLP metabolic stability curve was established by plotting the stopped time points (*x*-axis) from 0 to 70 min against the percentage remaining of SLP concentration compared to the zero-time concentration (*y*-axis) (Figure 4A). The linear part (0–30 min) of the constructed curve was selected to establish another natural logarithmic curve by establishing incubation time points (0–30 min) against the natural logarithm (Ln) of the % SLP remaining.

The parameters of SLP metabolic stability (in vitro t_1/2_ and the CL_int_) were calculated by the estimation of the % remaining SLP concentration after incubation with an active HLMs matrix that contains NADPH as a cofactor that starts the metabolic steps. The metabolic incubation steps were performed according to the following steps: first, pre-incubation of 1 µL of SLP (1 mM) with an active HLMs matrix without NADPH at 37 °C for 10 min so as to obtain the optimum parameters for enzymatic metabolic pathways. Second, starting the metabolic reaction was done by the addition of NADPH (1 mM) to the incubates. Third, stopping the metabolic reaction by the addition of 2 mL of ACN and, at the same time, adding 100 µL of FLG WK3 as the IS so as to avoid the metabolic effect on the FLG concentration. Stopping the incubation was done at certain time points: 0, 2.5, 7.5, 15, 20, 30, 40, 50, 60, and 70 min. The extraction procedure was conducted, as stated above. The first SLP metabolic stability curve was constructed by plotting the stopped time points (*x*-axis) from 0 to 70 min against the percentage of SLP concentration remaining compared to the zero-time concentration (*y*-axis). The linear part of the established curve was chosen to construct another natural logarithmic curve by establishing incubation time points against the natural logarithm (Ln) of the % SLP remaining. The slope of the second curve represented the rate constant for the SLP metabolic stability and was used to calculate the in vitro t_1/2_, whereas in vitro t_1/2_ = ln2/slope. Then, the SLP CL_int_ (µL/min/mg) was calculated [22], using a value of 45 mg of microsomal protein per gram of liver tissue and of 26 g for liver tissue per kilogram of body weight [30] (Equation 3).

## 4. Conclusions

An LC-MS/MS method was developed and validated for quantifying SLP in HLM’s matrix and applied for assessment of its metabolic stability. The LC-MS/MS method revealed optimum selectivity and sensitivity. The use of a small volume of ACN in the mobile phase composition makes the method eco-friendly. The established method is also characterized by high reproducibility of the recovery of analytes (FLG and SLP) from HLM’s matrix using the protein precipitation method. The in vitro metabolic experiment was performed to confirm the results of the in silico software. The outcomes of SLP metabolic stability [moderate CL_int_ (34 mL/min/kg) and in vitro t_1/2_ values (23.82 min)] revealed that SLP is a moderate clearance drug; so, a reasonable in vivo bioavailability could be proposed. From these results, we predicted that SLP could be administered to patients without the influence of dose accumulation inside the human body or rapid elimination by the liver. Future experiments may be done using this in silico and in vitro approach for developing new drugs with elevated metabolic stability behavior. The in vitro experiments of SLP matched the results of the in silico experiments, which reveals the importance of in silico metabolic stability studies to save time and effort. The developed LC-MS/MS method could be applied for therapeutic pharmacokinetic studies or drug monitoring (TDM) for SLP after readjusting the extraction steps using the same chromatographic parameters.

## Figures and Tables

**Figure 1 molecules-28-02618-f001:**
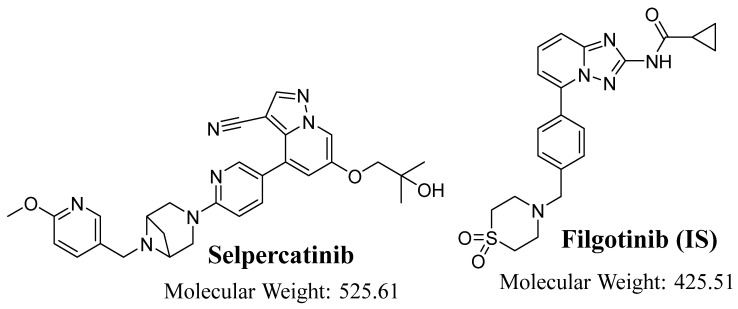
Chemical structures of selpercatinib and filgotinib (internal standard; IS).

**Figure 2 molecules-28-02618-f002:**
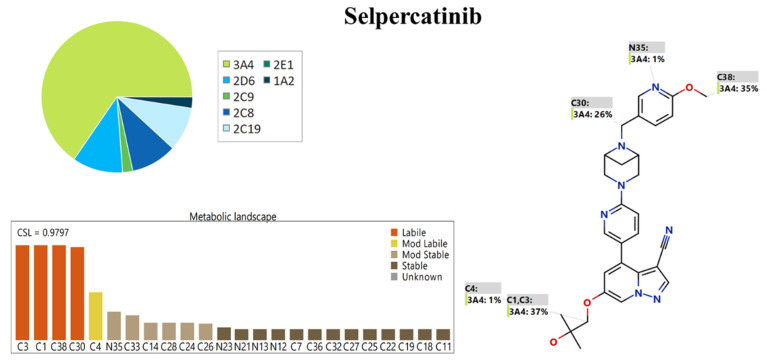
Metabolic landscape of SLP showing CSL (0.9797) and the proposed high SLP metabolic rate.

**Figure 3 molecules-28-02618-f003:**
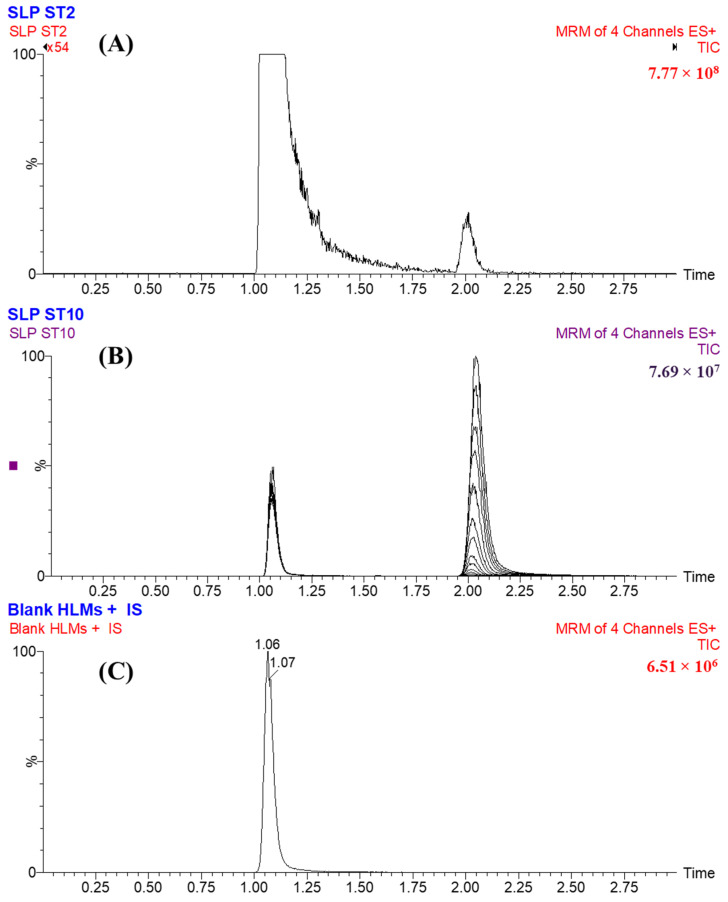
SLP LLQC at 1 ng/mL exhibiting the LC-MS/MS sensitivity (**A**) MRM chromatograms of the SLP calibration standards (**B**) exhibiting the SLP peak (2.1 min) and FLG peak (1.1 min) and MRM chromatograms of blank HLMs plus FLG (**C**) showing no carryover effect from SLP.

**Figure 4 molecules-28-02618-f004:**
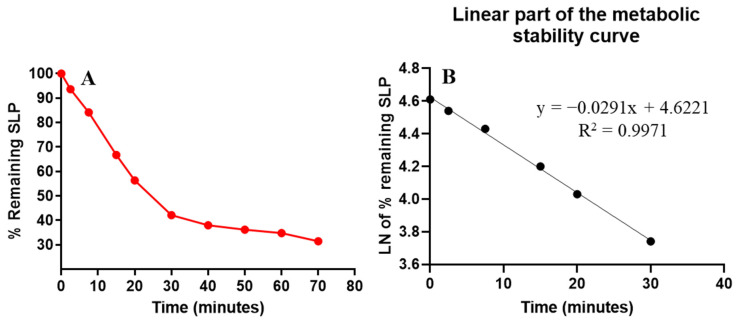
The curve of SLP metabolic stability in HLMs (**A**) and the linear LN calibration curve showing the linear regression equation with coefficient of variation (**B**).

**Figure 5 molecules-28-02618-f005:**
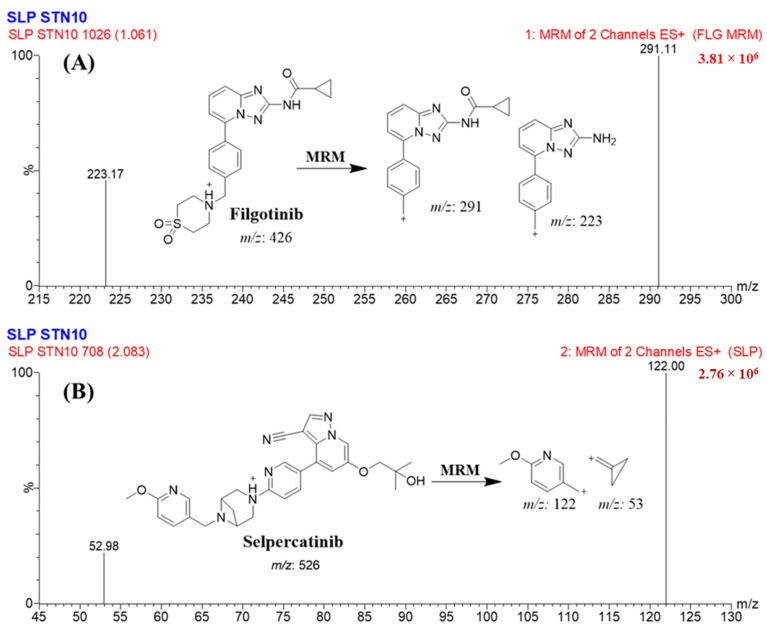
FLG MRM mass spectra. (**A**) SLP MRM mass spectra. (**B**) Exhibiting the proposed fragmentation pattern.

**Table 1 molecules-28-02618-t001:** Back-calculation results of six replicates (eleven standards) of SLP.

SLP Nominal Concentrations (ng/mL)	Mean	SD	RSD (%)	Accuracy (%)	Recovery
1 (LLQC)	0.95	0.02	2.08	−5.08	94.92
3 (LQC)	3.09	0.03	1.00	3.15	103.15
15	15.16	0.19	1.28	1.07	101.07
50	50.98	1.10	2.15	1.97	101.97
100	105.06	1.47	1.40	5.06	105.06
300	303.45	2.36	0.78	1.15	101.15
500	502.91	1.41	0.28	0.58	100.58
900 (MQC)	898.67	4.33	0.48	−0.15	99.85
1500	1492.58	11.37	0.76	−0.49	99.51
2400 (HQC)	2387.68	34.32	1.44	−0.51	99.49
3000	3025.03	36.16	1.20	0.83	100.83

**Table 2 molecules-28-02618-t002:** Precision and accuracy (intra- and inter-batch) results of the LC-MS/MS method.

SLP in HLM’s Matrix (ng/mL)	Intra-Batch Assay (Twelve Replicates in the Same Day)				Inter-Batch Assay (Six Replicates in Three Consecutive Days)			
	1 (LLQC)	3 (LQC)	900 (MQC)	2400 (HQC)	1 (LLQC)	3 (LQC)	900 (MQC)	2400 (HQC)
Mean	0.95	3.09	898.67	2387.68	0.93	3.16	912.08	2375.59
SD	0.02	0.03	4.33	34.32	0.02	0.10	15.59	24.66
Precision (%RSD)	2.08	1.00	0.48	1.44	2.01	3.08	1.71	1.04
% Accuracy	−5.08	3.15	−0.15	−0.51	−6.56	5.22	1.34	−1.02
Recovery (%)	94.92	103.15	99.85	99.49	93.44	105.22	101.34	98.98

**Table 3 molecules-28-02618-t003:** SLP metabolic stability.

Time (min)	Mean ^a^ (ng/mL)	X ^b^	LN X	Linearity Parameters
0	626	100.00	4.61	Regression equation: y = −0.0291x + 4.6221
2.5	586	93.61	4.54	
7.5	527	84.19	4.43	R^2^ = 0.9971
15	418	66.77	4.20	
20	353	56.39	4.03	Slope: −0.0291
30	264	42.17	3.74	
40	238	38.02	3.64	t_1/2_: 23.82 min and
50	227	36.26	3.59	Cl_int_: 34 mL/min/kg
60	218	34.82	3.55	
70	197	31.47	3.45	

Notes: ^a^ Average of three repeats. ^b^ X: Mean of the SLP % remaining of the three repeats.

**Table 4 molecules-28-02618-t004:** Various experiments for resolution of SLP and FLG peaks.

Analytes	Methanol	ACN	Solid Phase Extraction	Protein Precipitation	C18 or C8 Column	PFP Column
SLP	1.71 min	2.1 min	Low recovery	High recovery	2.65 min	2.1 min
	Tailed	Perfect	Irreproducible	Reproducible	Tailed	Perfect
FLG	1.34 min	1.1 min	Low recovery	High recovery	2.34 min	1.1 min
	Overlapped	Perfect	Irreproducible	Reproducible	Perfect	Perfect

**Table 5 molecules-28-02618-t005:** MRM parameters for the quantification of SLP and FLG (IS).

Drug	ESI Mode	Rt	Precursor (*m/z*)	Qualification Traces (*m/z*)	Quantification Traces (*m/z*)	Collision Energy (CE, eV)	Cone Voltage (V)
SLP	+ve	2.1	526.0	52.98	122.0	66/28	42
FLG (IS)	+ve	1.1	426.0	223.17	291.11	24/38	38

## Data Availability

All data are available within the manuscript.

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
