# Peer review of "Assessment of In Silico and In Vitro Selpercatinib Metabolic Stability in Human Liver Microsomes Using a Validated LC-MS/MS Method"

_molecules, 2023, doi:10.3390/molecules28062618_

Round 1
Reviewer 1 Report
The authors Attwa et al. describe the development of an LC-MS/MS based method for quantification of Selpercatinib (SLP), a rearranged during transfection (RET) inhibitor, in human liver microsomes (HLM). The drug was recently approved by FDA for adult patients with locally advanced or metastatic solid tumors with RET gene fusion. The authors have also applied the developed method for the metabolic stability assessment of the drug. The study is highly significant. However, following points need to be addressed before the publication of this work:
1. References are not in order. Reference 10 is listed after reference 11. Please make sure all the reference are listed in the order that they appear in the text.
2. Page 3, Lines 85-88 – “Another advantage of the current study--------------------reported methods previously [9]” Both reference 8 and 9 should be cited here?
3. Page 4, Lines 136-139 – Please adjust Figures 3 (B) and Figure 3 (C) in the order they appear in the text. The text discusses Figure 3 (C) before Figure 3 (B).
4. Please specify the LC column temperature and autosampler temperature under LC conditions.
5. Authors briefly discuss the optimization of LC conditions, but it would be helpful for the readers if the generated data (such as representative chromatograms) is shared in the supporting information.
6. Specify the collision energy used for MRM. Please discuss if there was any optimization performed for selecting the suitable collision energy?
Author Response
Dear Mr.Hyland Xue, Assistant Editor,
MDPI Nanjing, Molecules Journal
Manuscript ID: molecules-2248821
Kind regards,
“Editor Comments”
(I) Please revise your manuscript according to the referees’ comments and
upload the revised file within 5 days.
(II) Please use the version of your manuscript found at the above link for
your revisions.
(III) Please check that all references are relevant to the contents of the
manuscript.
(IV) Any revisions made to the manuscript should be marked up using the
“Track Changes” function if you are using MS Word/LaTeX, such that
changes can be easily viewed by the editors and reviewers.
(V) Please provide a short cover letter detailing your changes for the
editors’ and referees’ approval.
If one of the referees has suggested that your manuscript should undergo
extensive English revisions, please address this issue during revision. We
propose that you use one of the editing services listed at
https://www.mdpi.com/authors/english or have your manuscript checked by a
native English-speaking colleague.
Please do not hesitate to contact us if you have any questions regarding the
revision of your manuscript or if you need more time. We look forward to
hearing from you soon.
Authors’ response
We thank the editor this opportunity to improve our manuscript and be considered again for publication in Molecules Journal. We give below detailed answers to each question raised by reviewer # 1. All replies to the comments were highlighted in yellow color in the revised manuscript.
Reviewer # 1
Comments and Suggestions for Authors
The authors Attwa et al. describe the development of an LC-MS/MS based method for the quantification of Selpercatinib (SLP), a rearranged during transfection (RET) inhibitor, in human liver microsomes (HLM). The drug was recently approved by FDA for adult patients with locally advanced or metastatic solid tumors with RET gene fusion. The authors have also applied the developed method for the metabolic stability assessment of the drug. The study is highly significant. However, following points need to be addressed before the publication of this work:
Authors’ response
We appreciate the reviewer’s words and his/her suggestions to improve our manuscript. We give below our answer to his/her concerns.
Point # 1
- References are not in order. Reference 10 is listed after reference 11. Please make sure all the reference are listed in the order that they appear in the text.
All references were rechecked. The typo mistake was corrected. Reference 11 was changed to reference 8 as it is one of the previously published article for SLP quantification.
Point # 2
- Page 3, Lines 85-88 – “Another advantage of the current study--------------------reported methods previously [9]” Both reference 8 and 9 should be cited here?
The references were cited as requested.
The current LC-MS/MS method characterized by the using of isocratic mobile phase in short run time (3 min) and less flow rate (0.4 mL) that is considered more precise and consuming less organic solvent (green chemistry) in comparison with the reported meth-od [8]. Furthermore, the constructed calibration curve showed a linearity in the range of 1 to 3000 ng/mL that revealed wider range than the previous reported calibration model [8]. Another vantage of the current study is quantifying of SLP using MRM transitions (m/z 525 → 122 and 525 → 59) that is more accurate and sensitive comparing to SRM transition (m/z 525 → 122) in the other two reported methods previously [8, 9]. The current work is considered the first analytical method to evaluate SLP metabolic stability in HLMs.
Point # 3
- Page 4, Lines 136-139 – Please adjust Figures 3 (B) and Figure 3 (C) in the order they appear in the text. The text discusses Figure 3 (C) before Figure 3 (B).
The changes were performed as requested.
Figure 3B illustrates the reproducibility of the LC-MS/MS method by showing the MRM chromatograms of SLP calibration levels (1 ng/mL to 3000 ng/mL) that revealed the same retention times for all calibration standards of SLP. No observed carry-over was found for SLP in blank HLMs plus FLG MRM chromatogram (Figure 3C).
Point # 4
- Please specify the LC column temperature and autosampler temperature under LC conditions.
The requested information were added:
The column temperature was kept at room temperature at 23 ± 1 °C. The auto sampler temperature was kept 5 °C to increase the stability of samples.
Point # 5
- Authors briefly discuss the optimization of LC conditions, but it would be helpful for the readers if the generated data (such as representative chromatograms) is shared in the supporting information.
More information was added as requested:
All data were updated in the manuscript.
Various trials were done to choose the perfect chromatographic condition for resolution, extraction and analysis of SLP and FLG analytes peaks in a good shape and in a short retention time as mentioned in table 2.
Table 2. Different experiments for separation of analytes peaks.
|
Analytes |
Methanol |
ACN |
Solid phase extraction |
Protein precipitation |
C18 or C8 column |
PFP column |
|
SLP |
1.71 min |
2.1 min |
Low recovery |
High recovery |
2.65 min |
2.1 min |
|
Tailed |
Perfect |
Irreproducible |
Reproducible |
Tailed |
Perfect |
|
|
FLG |
1.34 min |
1.1 min |
Low recovery |
High recovery |
2.34 min |
1.1 min |
|
Overlapped |
Perfect |
Irreproducible |
Reproducible |
Perfect |
Perfect |
Point # 6
- Specify the collision energy used for MRM. Please discuss if there was any optimization performed for selecting the suitable collision energy?
The tuning for SLP and FLG MRM parameters were performed automatically using IntelliStart® software, which were fixed manually in fluidics and LC (combined mode) so as to elevate intensity and selectivity of the chromatographic peaks of the two analytes (SLP and FLG).
FLG and SLP chromatographic peaks were eluted at Rt: 1.1 min. (Figure 5A) and Rt: 2.1 min. (Figure 5B), respectively that were quantified through the use of MRM mode transitions. All MRM optimized parameters of SLP and FLG (IS) using IntelliStart® software are listed in Table 6.
Table 6. MRM parameters for the estimation of the two analytes (SLP and FLG (IS)).
|
Cone Voltage |
Collision Energy |
Quantification Traces (m/z) |
Qualification Traces (m/z) |
Precursor |
Rt |
ESI Mode |
Drug |
|
42 |
66/28 |
122.0 |
52.98 |
526.0 |
2.1 |
+ve |
SLP |
|
38 |
24/38 |
291.11 |
223.17 |
426.0 |
1.1 |
+ve |
FLG (IS) |
Kind regards
Dr. Mohamed Attwa
Dept. Pharm. Chem., College of Pharmacy, King Saud University,
P.O. Box 2457,
Riyadh, 11451, Saudi Arabia
mzeidan@ksu.edu.sa

Reviewer 2 Report
In this manuscript, the authors descript a LC-MS/MS method for the assessment of in silico and in vitro Selpercatinib (RET kinase inhibitor) metabolic stability in human liver microsomes. Multiple reaction monitoring (MRM) was used for quantification with filgotinib as internal standard and the method was validated in linearity, precision, accuracy, selectivity, lower limit of quantitation, matrix effects, stability, et al. Overall, the manuscript is well-written to the point and is good for publication with the following minor concerns:
1. What is the peak of blank run in fig 3C? It has similar retention time as FLG?
2. A bit more work need to be done on English proof reading.
Author Response
Dear Mr.Hyland Xue, Assistant Editor,
MDPI Nanjing, Molecules Journal
Manuscript ID: molecules-2248821
Kind regards,
“Editor Comments”
(I) Please revise your manuscript according to the referees’ comments and
upload the revised file within 5 days.
(II) Please use the version of your manuscript found at the above link for
your revisions.
(III) Please check that all references are relevant to the contents of the
manuscript.
(IV) Any revisions made to the manuscript should be marked up using the
“Track Changes” function if you are using MS Word/LaTeX, such that
changes can be easily viewed by the editors and reviewers.
(V) Please provide a short cover letter detailing your changes for the
editors’ and referees’ approval.
If one of the referees has suggested that your manuscript should undergo
extensive English revisions, please address this issue during revision. We
propose that you use one of the editing services listed at
https://www.mdpi.com/authors/english or have your manuscript checked by a
native English-speaking colleague.
Please do not hesitate to contact us if you have any questions regarding the
revision of your manuscript or if you need more time. We look forward to
hearing from you soon.
Authors’ response
We thank the editor this opportunity to improve our manuscript and be considered again for publication in Molecules Journal. We give below detailed answers to each question raised by reviewer # 2. All replies to the comments were highlighted in yellow color in the revised manuscript.
Reviewer # 2
Comments and Suggestions for Authors
In this manuscript, the authors descript a LC-MS/MS method for the assessment of in silico and in vitro Selpercatinib (RET kinase inhibitor) metabolic stability in human liver microsomes. Multiple reaction monitoring (MRM) was used for quantification with filgotinib as internal standard and the method was validated in linearity, precision, accuracy, selectivity, lower limit of quantitation, matrix effects, stability, et al. Overall, the manuscript is well-written to the point and is good for publication with the following minor concerns:
Authors’ response
We appreciate the reviewer’s words and his/her suggestions to improve our manuscript. We give below our answer to his/her concerns.
Point # 1
- What is the peak of blank run in fig 3C? It has similar retention time as FLG?
Fig 3C represented MRM chromatogram of positive blank HLMs plus FLG (C) showing no carry over effect from SLP.
Point # 2
- A bit more work need to be done on English proof reading.
The manuscript has been edited to ensure language and grammar accuracy by Editage (a brand of Cactus Communication). We attached a certificate of English editing.
Kind regards
Dr. Mohamed Attwa
Dept. Pharm. Chem., College of Pharmacy, King Saud University,
P.O. Box 2457,
Riyadh, 11451, Saudi Arabia
mzeidan@ksu.edu.sa
